



# Risk-based analysis of monitoring time intervals for landslide prevention

Jongook Lee[1], Dong Kun Lee[2], Sung-Ho Kil[3], Ho Gul Kim[4]

[1]Interdisciplinary Program in Landscape Architecture, Seoul National University, Seoul, 08826, Republic of Korea
[2]Research Institute of Agriculture Life Science, Seoul National University, Seoul, 08826, Republic of Korea
[3]Department of Ecological Landscape Architecture Design, Kangwon National University, Chuncheon, 24341, Republic of Korea
[4]Urban Data and Information Center, Incheon Development Institute, Incheon, 22711, Republic of Korea

*Correspondence to*: Dong Kun Lee (dklee7@snu.ac.kr)

**Abstract.**

Landslide is one of the most dangerous disasters in terms of occurrence frequency and damage severity that result in loss of human life and social infrastructure. Monitoring methods based on low temporal resolution instruments such as an inclinometer or piezometer can be an effective and cost-efficient solution. The objective of this research is to analyse monitoring time intervals for low temporal resolution methods based on a risk study and to propose a plan for periodic landslide monitoring along with the different landslide hazard areas by considering the risk reduction effect. For this purpose, an equation for the probability of landslide occurrence was denoted by the concept of reliability, and the monitoring time interval was analysed quantitatively by calculating the average probability of landslide occurrence. To identity the frequency of landslide occurrence, a unit of relative temporal frequencies was adopted, and it was estimated by establishing rainfall threshold. As a case study site, Pyeongchang County was selected, where landslide inventory data are available and an increase in population and infrastructure has been observed since Pyeongchang became the host city of the 2018 winter Olympic games. The result demonstrates that the appropriate monitoring intervals can be determined by calculating the average probability of landslide occurrence, and resources can then be allocated efficiently for landslide prevention.

# 1 Introduction

Climate change such as an increase in heavy precipitation events has been observed recently, and the risk to people, properties and ecosystems from natural disasters such as storms, extreme precipitation and landslides has been intensified, especially under conditions of insufficient infrastructure and services (IPCC, 2013). Landslide is one of the most dangerous disasters in terms of occurrence frequency and damage severity that result in loss of human life and social infrastructure (Petley, 2012;Kjekstad and Highland, 2009;Guzzetti, 2000;Salvati et al., 2010).



However, almost all countries find it challenging to have enough financial and human resources for major engineering works or relatively expensive monitoring methods to control disasters with installation of costly measures to manage risk from landslides (Guzzetti et al., 1999;Baroň and Supper, 2013). In fact, monitoring and warning systems can substitute for expensive stabilization works or engineering solutions to prevent loss from landslides wherever a large extent of area needs to be applied

(Dai et al., 2002). Monitoring methods based on low temporal resolution instruments, such as a conventional inclinometer to detect subsurface deformation and a piezometer to monitor pore water pressure, can be effective solutions (Uhlemann et al., 2016), and those methods can be useful for controlling multiple areas susceptible to landslides, where those areas are scattered among relatively numerous locations over a broad region. Nonetheless, a continuous monitoring example of landslide triggering parameters induced from rainfall is occasionally sought due to time and cost restrictions (Springman et al., 2013),

and a few articles have addressed the time intervals for monitoring to control unstable slope hazards with risk-based analysis. In the case of the Republic of Korea, the monitoring frequency of the unstable slopes is regulated by law, simply requiring that they be monitored more than once in a year. The objective of this research is to analyse monitoring time intervals for low temporal resolution methods based on a risk study and to propose a plan for periodic landslide monitoring along with the different landslide hazard areas considering risk reduction effects.

As a case study site, Pyeongchang County was selected where landslide inventory data are available after Typhoon Ewiniar caused strong rainfall and resulted in the loss of lives and property in July 2006; recently in the county, the increase in population and expansion of infrastructure have been observed since Pyeongchang became the host city of the 2018 winter Olympic games. To reflect the spatial probability data from landslide susceptibility classification, the landslide hazard map produced by the Korea Forest Service was used (Korea Forest Service, 2012), and temporal probability was incorporated into

this study by considering the frequency of landslide occurrence estimated from the landslide inventory data.

For estimation of the frequency of landslide occurrence, a descriptor of relative temporal frequencies was adopted as suggested by (Corominas and Moya, 2008) to conduct the study on a regional scale, and the number of landslide events was analysed along with the landslide hazard grades. Despite the limitation on the accumulated landslide inventory data over the study region, the frequency of landslide occurrence was estimated by establishing a rainfall threshold. The selection of the

rainfall threshold has been examined and proposed by various methodologies (Crozier, 2005;Guzzetti et al., 2007;Zêzere et al., 2005;Frattini et al., 2009;Crosta and Frattini, 2003;Aleotti, 2004;Polemio and Sdao, 1999;Caine, 1980), and it has been shown that proposed rainfall thresholds are applicable to the respective study cases depending on the landside type with local validity (Jakob et al., 2006;Martelloni et al., 2012), so the criteria of cumulative precipitation and daily rainfall intensity for rainfall threshold setting were determined by referring to the local research results.

To denote the probability of landslide occurrence, the concept of reliability is used to derive the formula for the probability of landslide occurrence in this study, and its outcome is the same as the equation expressed by (Crovelli, 2000) based on a Poisson distribution model that is frequently employed to demonstrate the random events of natural hazards on a continuous timeline. The monitoring time interval is calculated quantitatively to reduce the landslide risk by adopting the concept of time-



average probability of failure that is often used in reliability studies for safety instrumented systems (I.S.A., 2002) and has also been applied to structural safety for earth dams (Su et al., 2012).

This paper demonstrates that differentiated monitoring time intervals need to be scheduled along with discrete landslide hazard areas in order to achieve the same level of risk reduction effect, and the result can be a useful basis upon which to

improve preventive landslide risk management with monitoring activities.

## 2 Method

### 2.1 Study site

Pyeongchang is well known as the host city of the 2018 Olympic Winter Games, and most of the snow sports competitions

will take place at the nearby ski resorts. The county of Pyeongchang is located between 37°15' and 37°50' north-south latitudes and 128°15' and 128°45' west-east longitudes near the mountain range in Gangwon Province of South Korea as displayed in Fig. 1. The total area of Pyeongchang is 1465 km$^2$, which accounts for 8.7% of the province area. Its elevation is generally high, with an average altitude of more than 600 m above sea level, and the northeast area is higher and includes steeper terrain than the southwest side. This particular topographical feature makes the study site prone to landslide.

The agricultural products are mainly from high altitude cultivation of maize, potato, and cabbage, and some of the crop fields are located at the hilly side near the mountainous region. The population of Pyeongchang has been growing recently, and statistics show that the registered number of resident households has increased from 16251 in the year 2000 to 20745 in 2016 (Statistics Korea, 2017), which reflects the high demand for leisure and investment in the county. As a result, housing developments for permanent residences and for tourism are sprawling near the mountainous area where they are relatively

vulnerable to landslides.

Furthermore, it has been indicated from the previous study on the Gangwon province region that the landslide hazard area will be increased by climate change impacts in the future when estimated with Representative Concentration Pathways (RCP) scenarios (Kim et al., 2014). Thus, implementation of monitoring with a proper management plan is required in the county to prevent potential loss of lives and property that may occur as a result of landslides.




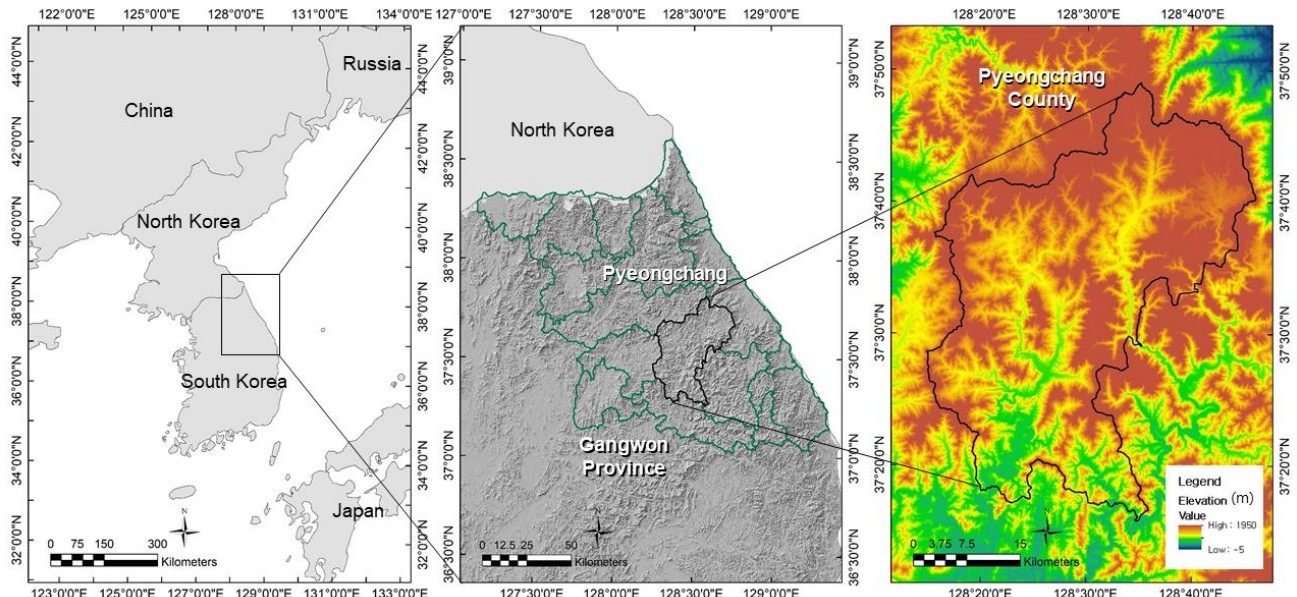

**Figure1.** The study site of Pyeongchang County in Gangwon province, South Korea

## 2.2 Landslide susceptibility data

To obtain the spatial probability data from landslide susceptibility classification, we used the landslide hazard map prepared in national scope by the Korea Forest Service in 2012, which was made based on logistic regression analysis (Korea Forest Service, 2012). The landslide hazard map was produced to be used as basic data for surveys of vulnerable areas to landslides and for implementation of preventive action.

    According to the map, the scope areas are classified into five classes from grade one (the highest hazard) to grade five (the

lowest hazard), where landslides can be triggered by strong rainfall on forests across the country. The nine factors of mountain characteristics were analysed to produce the hazard map, and those are slope inclination, slope orientation, slope length, slope curvature, topographic wetness index (TWI), the type of forest, the age of the forest, soil depth and bedrock. The data from the landslide hazard map were used in 1: 25000 scale, and the size of the pixel unit is 10 m×10 m.

    It should be noted that the lowest grade of landslide hazard data, grade five (5), was excluded in this study while analysing

landslide occurrence frequency because those pixel data were entered in the data set as null including waterbody and plain low land.



## 2.3 Landslide inventory data

The landslide inventory data were provided by the municipality of Gangwon province. The inventory was conducted after devastating damages were reported after Typhoon Ewiniar that brought strong rainfall to the region in July 2006. Ewiniar originated in the western North Pacific as a tropical depression, and it became a category four super typhoon on July 4 with a

minimum central pressure of 930 hPa (Zheng et al., 2008). The typhoon weakened into a tropical depression accompanied by torrential storm rain when it reached the Korean peninsula on July 10. However, it caused 62 casualties in total, and the loss of property was estimated to be approximately 1834.4 billion Korean won according to the report by the National Emergency Management Agency of South Korea (N.E.M.A., 2007; K.M.A.N.T.C., 2011).

The landslide inventory data are part of the first accumulation after the field survey, and a total of the 1751 landslide

locations were identified in Pyeongchang County. The survey result was digitized in polygon type in the GIS platform, and it consists of location data positioned with GPS devices and the area data of damage by landslide scar. The inventory information of location and the area affected were plotted on a 1:25000 scale map.

Unfortunately, time-series inventory data after Ewiniar were not available; however, the frequency of landslide occurrence was estimated with this cross-sectional data by establishing a rainfall threshold, as described in the method below. These

inventory data do not include landslide types with hazard descriptors, which can distinguish most landslides and rock fall with four categories: rock fall from cliffs or cut-slope, cuts and fills on roads or railways, small landslide, and individual landslides on natural slopes (Fell et al., 2008). Despite the limitations of the inventory, the data provide a spatial distribution of 1751 landslide locations and indicate the overall alteration of the landslide occurrence ratio corresponding to landslide susceptibility variance, which are presented in the results section of this paper.

## 2.4 Probability of landslide occurrence

### 2.4.1 Probability model by Poisson distribution

Using landslide frequency data obtained from the past landslide inventory result, the temporal probability of landslide occurrence can be estimated by means of statistical analysis. There have been studies using the temporal probability of

landslide occurrence to generate hazard maps not only considering spatial probability that means the potential of slope terrain failure (Lopez Saez et al., 2012;Tien Bui et al., 2013;Guzzetti et al., 2005). To present the probability of landslide occurrence, Crovelli (2000) used a Poisson distribution model as below, which is frequently adopted to demonstrate random events in continuous time in natural environments. The probability of *n* landslides occurring during time *t* is:

$$P\{N(t) = n\} = e^{-\lambda t}\, \frac{(-\lambda t)^n}{n!} \qquad\qquad (1)$$





where n = 1, 2, 3 …,

$\lambda$ is the rate of occurrence of landslides,

$t$ is the specified time,

$N(t)$ is the number of landslides that occurred during time $t$.

The probability of a landslide recurrence in time interval $Ti$ that is greater than time $t$ is, which means if $\lambda$ is much less than one ($\lambda \ll 1$):

$$P(Ti > t) = P\{N(t) = 0\} = e^{-\lambda t} = e^{-t/\mu} \qquad (2)$$

where $Ti$ is the time between landslide events,

$\mu$ is the mean recurrence interval in the future.

Finally, the probability of one or more landslides occurring during time $t$, the so-called exceedance probability, is expressed

as:

$$P\{N(t) \geq 1\} = 1 - P\{N(t) = 0\} = 1 - e^{-\lambda t} = 1 - e^{-t/\mu} \qquad (3)$$

**2.4.2 Probability model by the reliability concept**

This probability of landslide occurrence model presented by the Poisson model can also be expressed using the concept of reliability. There are various definitions of reliability depending of the field of study; however, it contains four basic elements: probability, adequate performance, time, and operating conditions (Billinton and Allan, 1992). One of the definitions of reliability in general terms can be introduced as: the probability that an item will perform a required function without failure under stated conditions for a stated period of time (O'Connor and Kleyner, 2012). According to Kapur and
Pecht (2014), the reliability function $R(t)$ in mathematical terms is expressed as:

$$R(t) = \frac{Ns(t)}{N} \qquad (4)$$

where $Ns$ is the number of surviving items,

$N$ is the number of total items.



Unreliability, *F(t)* is given as:

$$F(t) = 1 - R(t) = \frac{N - Ns(t)}{N} \qquad (5)$$

$$\text{and, } f(t) = \frac{dF(t)}{dt} = -\frac{1}{N}\frac{dNs(t)}{dt} \qquad (6)$$

When the hazard rate *h(t)* is normalized with its surviving items *Ns(t)* instead of the total number of items *N* from the unreliability rate *f (t)* equation, we have the hazard rate *h(t)* as below with a more conservative meaning in terms of reliability.

$$h(t) = \frac{f(t)}{R(t)} \qquad (7)$$

The integral of the hazard rate *h(t)* over the time from 0 to *t* is:

$$\int_0^t h(\tau)\, d\tau = -\ln R(t) \qquad (8)$$

Then, *R(t)* is:

$$R(t) = e^{-H(t)} \qquad (9)$$

where *H(t)* is the number of hazards in time *t* and can be expressed as *H(t)*= λt.

Finally, we have *F(t)* as:

$$F(t) = 1 - R(t) = 1 - e^{-H(t)} = 1 - e^{-\lambda t} \qquad (10)$$

Which is same as Eq. (3) above for the probability of the landslide occurrence model.

## 2.5 Frequency of landslide occurrence

Landslide risk can be expressed with an equation that includes the probability of occurrence multiplied by the value of the
element at risk of landside, and to calculate the probability of landslide occurrence, the frequency of landslide occurrence should be estimated first (Corominas et al., 2014). However, landslide investigation data are not yet sufficiently accumulated





in the study region to estimate landslide occurrence frequency. The lack of landslide occurrence information and the difficulty of achieving a complete or systematic landslide record from the past have been demonstrated in previous research (Jaiswal et al., 2010;van Westen et al., 2006;Cardinali et al., 2002). From the previous studies, methods to estimate landslide frequency from the available information were summarized as five schemes: observation experience at a site, inventories from landslide

statistics, triggering event equating, cause and effect model from geomorphology, and deterministic stability model (A. G. S., 2000).

Furthermore, Corominas & Moya (2008) mentioned two separate approaches to spatial probability and temporal probability of landslide occurrence and differentiated the frequency of landslide with three descriptors: absolute frequency, relative frequency and indirect frequency, through their review research. Among those descriptors, the relative frequency was

given as a ratio of the number of landslides directly recorded to the unit area, which enables us to study regional multiple landslide events.  In this study, we have attempted to identify the relative temporal frequency of landslides because the available landslide inventory data include multiple landslides in the broad region triggered by an occasional intensive storm rain.

Provided that the unit pixel is from the data of a landslide hazard map that is a product of spatial probability analysis, the relative temporal frequency of landslides for Pyeongchang County is described quantitatively as below:

$$F_L = \text{landslide event} \times \text{pixel}^{-1} \times \text{year}^{-1} \qquad (11)$$

where $F_L$ is the relative temporal frequency of landslides, and the area of a unit pixel corresponds to 100 m$^2$, that is 10 m ×10 m unit pixel of the landslide hazard map.

This is based on the following assumptions.  To count the number of landslide events classified by the landslide hazard grade, we found the maximum landslide hazard grade value overlaid on the inventory polygon data using GIS and regarded it as the representative hazard grade for that landslide occurrence point. Second, to determine the total area corresponding to each landslide hazard grade, we considered the area where landslides do not occur at all, before it was used to arithmetically

divide the number of events to gain the frequency value in GIS. The reason for this is to approach the risk analysis conservatively by selecting a high failure rate. And also notably, the occurrence frequency varies depending on the size of the area. The frequency unit has a representative value only to the Pyeongchang administrative area of extent when its area is fully extended to cover the total county area, if the inventory has been completely executed and recorded in the entire Pyeongchang County area. Lastly, to estimate the probabilistic period of landslide occurrence with an equivalent magnitude, the

meteorological pattern of rainfall intensity and duration was considered as a triggering factor correlated with landslide occurrence. The rainfall threshold to initiate a landslide will be further explained below.

On the other hand, the relative temporal frequency of landslide can be derived from the concept of reliability as below:

$$F_L = (N - N_s) / (N_s * \Delta t) \qquad (12)$$



where $\Delta t$ is the probabilistic period of landslide occurrence.

Provided that $N - Ns$ is considered as the number of landslide events rather than the area because it is the concept of the
point source where the landslide initiated, the temporal landslide frequency expressed in Eq. (12) can be interpreted from the
definition of unreliability in Eq. (5), when $Ns$ is considered as the number of pixel components with no landslide initiation and
$\Delta t$ as the probabilistic period of landslide occurrence estimated by establishing the rainfall threshold.

## 2.6 Rainfall threshold triggering landslide

The mechanism of landslide occurrence potentially triggered by storm rain with high intensity and long duration can be
explained by the increase in pore water pressure and rain water seepage forces (Cullen et al., 2016). Since Caine's research
(1980) examined the relationship between the minimum rainfall duration and intensity required to cause a landslide, clarifying
the importance of precipitation duration and not only daily rainfall, and furthering the research, Guzzetti et al. (2008) updated
the rainfall intensity and duration values with a global database.

To determine the rainfall threshold for triggering a landslide, various methodologies have been examined and proposed
prior to assess landslide hazards (Crozier, 2005;Guzzetti et al., 2007;Zêzere et al., 2005;Frattini et al., 2009;Crosta and Frattini,
2003). In addition, it has been shown that an area that presents relatively high susceptibility to landslide could be a low hazard
area without intense enough rainfall above the threshold (Jaiswal et al., 2010). Based on the previous studies, we also estimated
the temporal probability of landslide occurrence by applying a rainfall threshold with daily rainfall level and duration that may
trigger landslides.

In determining the criteria of rainfall duration, the selection of the number of days from the event initiation is controversial.
Different numbers of days have been considered and examined by researchers to find the most suitable correlation with
landslide initiation (Aleotti, 2004;Polemio and Sdao, 1999;Zêzere et al., 2005). Of the issue, the researchers have proposed
different numbers of days, however, only with local validity, which means that suggested rainfall thresholds are applicable to
the respective study cases, and it also depends on the landside type (Jakob et al., 2006;Martelloni et al., 2012). Therefore, we
referred to the domestic research results in that characteristics of local geology, soil and vegetation, as well as precipitation
patterns are different from the areas of other studies.

According to the research result of Kim and Chae (2009), it has been reported that landslides show a tendency to occur
when there is rainfall of more than 200 mm for 48 h in Gyeonggi province of South Korea. On this basis, cumulative
precipitation of more than 200 mm for 48 h was adopted as one of the criteria to decide the rainfall threshold. To determine
the daily rainfall intensity, which is another factor affecting the threshold, the daily precipitation level was reviewed for July
2006 when Typhoon Ewiniar led to storm rain, assuming that landslide will occur in the future with a similar magnitude as in
the past.



Although there is always variability in rainfall, this is a method to estimate the frequency of landslide occurrence without continuous landslide inventory data, and it is possible to predict that the frequency of occurrence will meet the assumptions if the observation is conducted for a long enough period of time. In addition, there must be non-uniformity of rainfall intensity over the study area when the typhoon affects it, and the location where precipitation data are recorded does not match exactly with the area damaged by landslides; however, the weather data were adopted as a representative sample to estimate landslide frequency, which become input data to fulfil the main purpose of this study, examining the risk reduction effect by calculating the probability of landslide occurrence. Further study with accumulated landslide inventory data will be able to produce a more accurate estimation of the probability of landslide occurrence.

## 2.7 Average probability of landslide occurrence

From the probability of landslide occurrence model either derived from the Poisson distribution or the reliability concept, it can be seen that the probability of initiating one or more landslides will increase over time. However, if we do risk management through periodic monitoring with a low temporal resolution method, it can also be assumed from the model that the probability of landslide occurrence will be reduced.

In order to quantitatively estimate the monitoring interval needed to reduce the risk of landslide occurrence below the desired level, we adopted the concept of time-average probability of failure, which is commonly used in reliability studies. The concept of time-average probability of failure, used in the standard IEC 61511 by the International Electro-technical Commission and ISA-TR84 by the Instrumentation, Systems, and Automation Society, is the average probability of failure on demand (PFDavg); it represents the average probability of a safety-instrumented shutdown system not operating in its desired function in an unwanted emergency situation (I.S.A., 2002; I.E.C, 2003). Those standards are applicable to a high demand mode such as a basic process control system and a low demand mode for an emergency system in separate ways. In this study, we referred to the proposed module on low demand mode reflecting the nature of landslide occurrence with relatively low frequency. The average probability of failure on demand is expressed by the formula below, as described in ISA-TR84; it is obtained by integrating the probability of failure function from time 0 to the proof testing time, which is the test for recovering components of the emergency shutdown system, and dividing by the time interval (I.S.A., 2002).

$$\text{PFDavg} = \frac{1}{\text{TI}} \int_0^{\text{TI}} 1 - e^{-\lambda t} \, dt \qquad (13)$$

where *TI* is the time interval between the proof tests, and $\lambda$ is the failure rate.

In this equation, if we consider the failure rate ($\lambda$) as a frequency of landslide occurrence and replace the test time interval with the monitoring interval to take action to prevent landslides or to avoid unwanted loss, the average probability of failure on demand can mean the time-average probability of landslide occurrence that is dependent on the monitoring time interval.



In other words, the formula shows that the probability of landslide occurrence can be managed at a certain level with stability through periodic monitoring in ideal conditions.

Although it is not achievable in reality due to the imperfectness of monitoring, an increase in the probability of occurrence can be expected up to a specific portion by periodic monitoring, and different constants will contribute to the probability equation by increasing the probability at each time of monitoring iteration. The increasing probability depends on the feature of the monitoring itself, the characteristics of the person performing it, or the nature of the topography, geology, soil structure and vegetation cover of the area. Despite the imperfectness of monitoring, it is possible to identify the effectiveness of monitoring with different time intervals assuming that periodic monitoring will manage the probability of occurrence in a steady state, and investigating and reflecting the constant value attribute to increase the probability of occurrence with monitoring activities can be a subject for further research.

Notably, in terms of structural safety for dams, Su et al. (2012) presented the differences in time-average probability of system failure by reinforcement interval in the search for an optimized earth dam reinforcement strategy, which is an example of applying the average probability of failure for an earthen structure collapsing. By using the time-average probability of failure, monitoring intervals were differentiated expecting the same level of risk reduction to the areas with discrete grades of landslide hazard in this study to suggest efficient landslide risk management.

## 2.8 Landslide monitoring on low temporal resolution

Monitoring is one of the key activities to prevent landslides, which can cause loss of lives and properties. Adequate monitoring to identify the kinematic and hydrological aspects together with a climatic parameter can assist in providing an alert before mass movements occur and suggesting a plan to manage risk to people who may be affected (Angeli et al., 2000). To acquire complete information regarding landslide triggering factors and mechanisms, various combinations of monitoring techniques with instruments are usually employed (Dixon et al., 2015). However, pore-water pressure, displacement and deformation were mentioned as the crucial parameters for landslide monitoring and early warning in previous studies (Baroň and Supper, 2013;Uhlemann et al., 2016).

Among the ground-based monitoring techniques, the conventional inclinometer to detect subsurface deformation and the Piezometer to monitor ground water have low temporal resolution, and monitoring methods based on low temporal resolution can be an effective solution to control the areas suspected of landslides (Uhlemann et al., 2016). In this study, we considered a landslide monitoring method with low temporal resolution to be limited to the analysis monitoring interval that a municipality can conduct with effective monitoring at low cost.



## 3 Result

### 3.1 Rainfall threshold triggering landslides

To determine the rainfall threshold triggering landslides as per the aforementioned method established, we reviewed the past precipitation data in the Pyeongchang County region. The survey results of daily rainfall and 48-hour cumulative rainfall in the region for the last 11 years, which were acquired from the data of the Automatic Weather Station (AWS) located in Pyeongchang County (K.M.A., 2017), are plotted in Fig. 2a and Fig. 2b below.

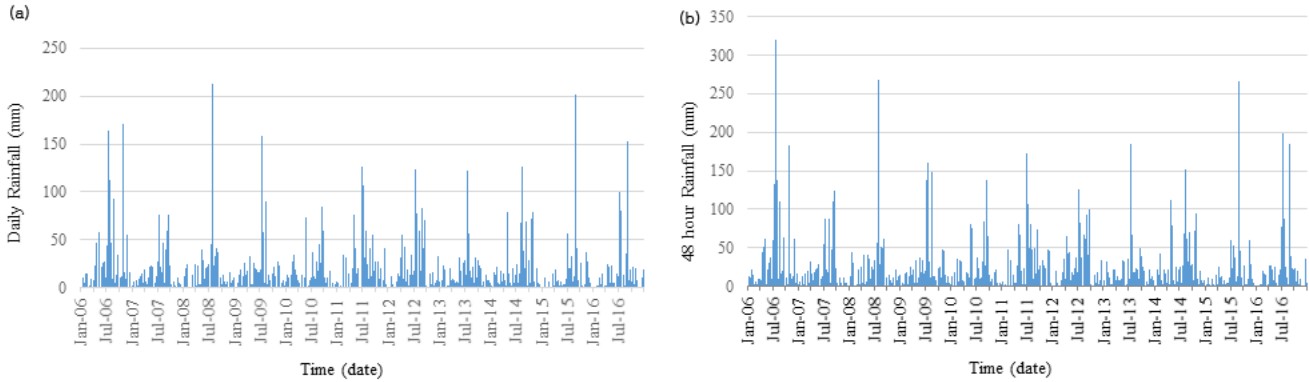

**Figure 2.** (a) Daily rainfall in Pyeongchang County for 11 years from 2006. The graph depicts the pattern of annual rainfall.
(b) 48-hour cumulative rainfall in Pyeongchang County. The graph displays 48-hour rainfall for 11 years from 2006.

The daily rainfall graph depicts that the precipitation pattern is characterized by the monsoon climate in which rainfall is mainly concentrated in summer, and intensive rainfalls are displayed on multiple occasions (K.M.A., 2017). Additionally, the graph shows that daily precipitation exceeded 150 mm on both July 15 and July 16, 2006, when the landslides were supposed to have occurred because of storm rain. Thus, a daily rainfall intensity of more than 150 mm was set as the rainfall threshold. As a result, the average landslide occurrence interval was estimated by counting the dates that meet both the 48-hour cumulative precipitation over 200 mm and the daily precipitation over 150 mm for 11 years from the year of 2006. In addition, it was identified as shown in Fig. 3 that three occasions in the last 11 years (July 16th in 2006, July 24th in 2008, Aug 25th in 2015) exceeded the rainfall threshold of this study, so the probabilistic period of landslide occurrence was estimated as 3.7 years.



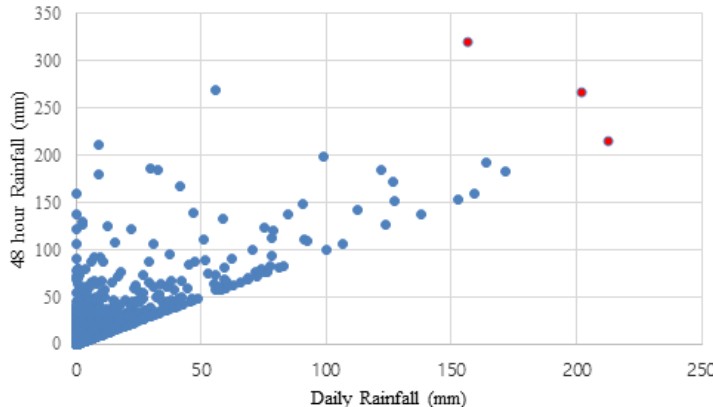

**Figure 3.** Scatter diagram between daily rainfall and 48-h cumulative rainfall in Pyeongchang County. The graph depicts the number of days that exceeded the rainfall threshold in the 11 years from 2006.

### 3.2 Landslide occurrence frequency

The landslide hazard map and locations where landslides have occurred in the inventory data are presented in Fig. 4 over the study area. The landslide hazard map of Pyeongchang County (Fig. 4a) shows that grade three (3) and grade two (2) are generally distributed over the study area and that grade one (1), the most hazardous one, is sparsely scattered along the ridges of the mountains. Meanwhile, the landslide locations drawn from the inventory data (Fig. 4b) reveal that the locations of landslide occurred primarily in the mountainous area in the north-eastern part of Pyeongchang County.

To investigate how the frequency of landslide occurrence varies according to landslide hazard grades, the landslide locations from the inventory data were overlaid, and a number of events were classified along with the grades. As shown in Table 1, the total areas of each grade were calculated, and the table indicates that grade three (3) occupies the largest area of 733.4 km$^2$, after that grade two (2) spread over the second largest area of 431.7 km$^2$, grade four (4) distributed on 24.9 km$^2$, and grade one (1) placed in the smallest area of 14.3 km$^2$. When counting the number of landslide events that took place at each landslide

hazard grade separately using GIS, a total of 111 landslides were found that belong to grade one (1), followed by 938 landslides that belong to grade two (2), 446 landslides that belong to grade three (3), and 5 landslides that belong to grade four (4). This figure indicates that the largest number of landslides occurred in the second grade area due to the total area it occupies; however, if we examine the occurrence ratio as presented in Table 1, which denotes the number of landslide events divided by total area, it is possible to identify that the highest number of landslide events per area has occurred in the first grade area.



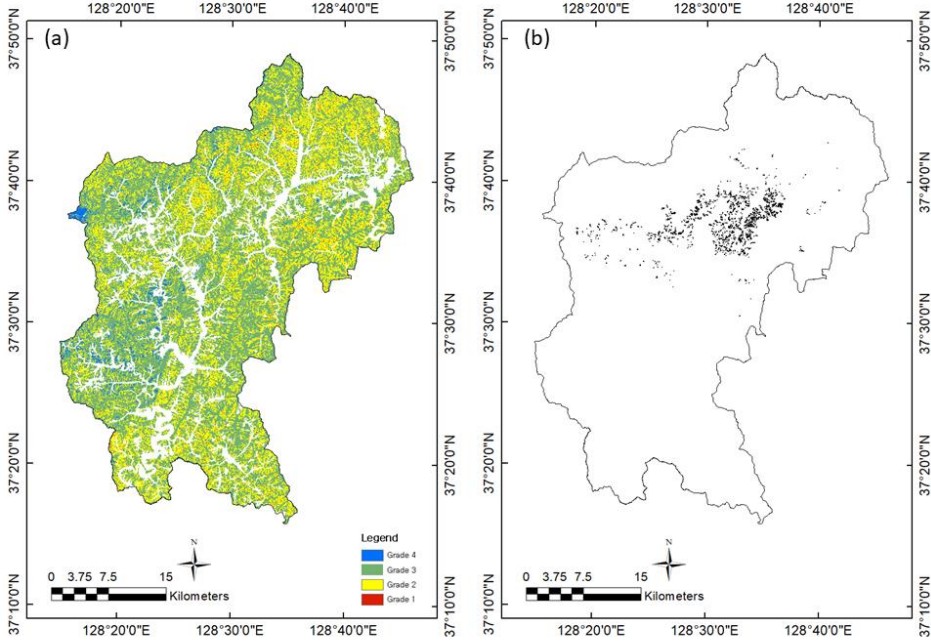

**Figure 4.** (a) Landslide hazard grade of the study area, (b) Locations of landslide occurrence in the study area

Next, using the unit of relative temporal frequency of landslides, the resulting landslide occurrence frequency ($\lambda$) for
5   Pyeongchang County was calculated as exhibited in Table 1. The result reveals that the area of landslide hazard grade one
(1) has the highest value of landslide occurrence frequency, which is 2.59E-05 (landslide event $\times$ pixel$^{-1}$ $\times$ year$^{-1}$), and a
sequentially decreasing value of the frequency from the second grade to the fourth grade is presented. In particular, it should
be noted that the numerical value of landslide occurrence frequency calculated as a result is variable depending on the unit
area. For instance, the given frequency value ($\lambda$) in Table 1 will be increased if we consider a larger unit area, such as a unit
10   of km$^2$ or to the whole county area. This issue will be examined further in the discussion section.





**Table 1.** The calculation result of landslide occurrence frequency

| Landslide hazard grade | No. of Landslides (N) | Total area * (A) | Occurrence Ratio (N/A) | Landslide occurrence frequency ** ($\lambda$) |
|---|---|---|---|---|
| 1 | 111 | 142761 | 7.78E-04 | 2.59E-05 |
| 2 | 938 | 4316864 | 2.17E-04 | 7.24E-06 |
| 3 | 446 | 7334461 | 6.08E-05 | 2.03E-06 |
| 4 | 5 | 248687 | 2.01E-05 | 6.70E-07 |

\* The unit of total area A is one pixel (10 m ×1 0m)

\*\* The unit of landslide occurrence rate ($\lambda$) is landslide event $\times$ pixel $^{-1}$ $\times$ year $^{-1}$

### 3.3 Landslide monitoring interval determination with average probability

By referring to the result of the landslide occurrence frequency calculated above, the graph in Fig. 5 on a logarithmic scale for the Y-axis delineates the change in average probability of landslide occurrence depending on the increasing monitoring interval, which is calculated based on the formula of average probability of landslide occurrence as Eq. (13). The logarithmic

plot in Fig. 5 categorizes discrete curves according to the landslide hazard grades, and it allows the analysis of differentiated monitoring time intervals for landslide prevention by considering risk reduction effects. The resulting graph illustrates that the highest average probability of landslide occurrence value of 1.06.E-04 is obtained for the first grade, and the probability value is gradually lowered to 2.96.E-05 for the second grade, to 8.29.E-06 for the third grade, and to 2.74.E-06 for the fourth grade, if we conduct the monitoring in the concerned areas with the same time interval, once per year. In other words, it

reveals that landslide monitoring with the same time interval for the different grades gives higher risk reduction effects to the lower hazard grade area than to the higher hazard grade area.

It is possible to identify through the graph that the same level of risk reduction effect can be expected regarding the second grade even if the monitoring is performed at an interval of 3.6 years, equivalent to that when the monitoring is conducted annually for the first grade. This analysis also technically demonstrates that the risk reduction effect that is the

same as anticipated from the landslide monitoring activities conducted annually for the first grade area is achievable even though the monitoring activities for the third grade sites and the fourth grade sites are planned with periods of 12.8 years and 38.7 years, respectively, on a rotational basis. Table 2 shows the discrete values of the average probability of landslide occurrence producing the same level of risk reduction effect with monitoring time interval variation according to the landslide hazard grades.






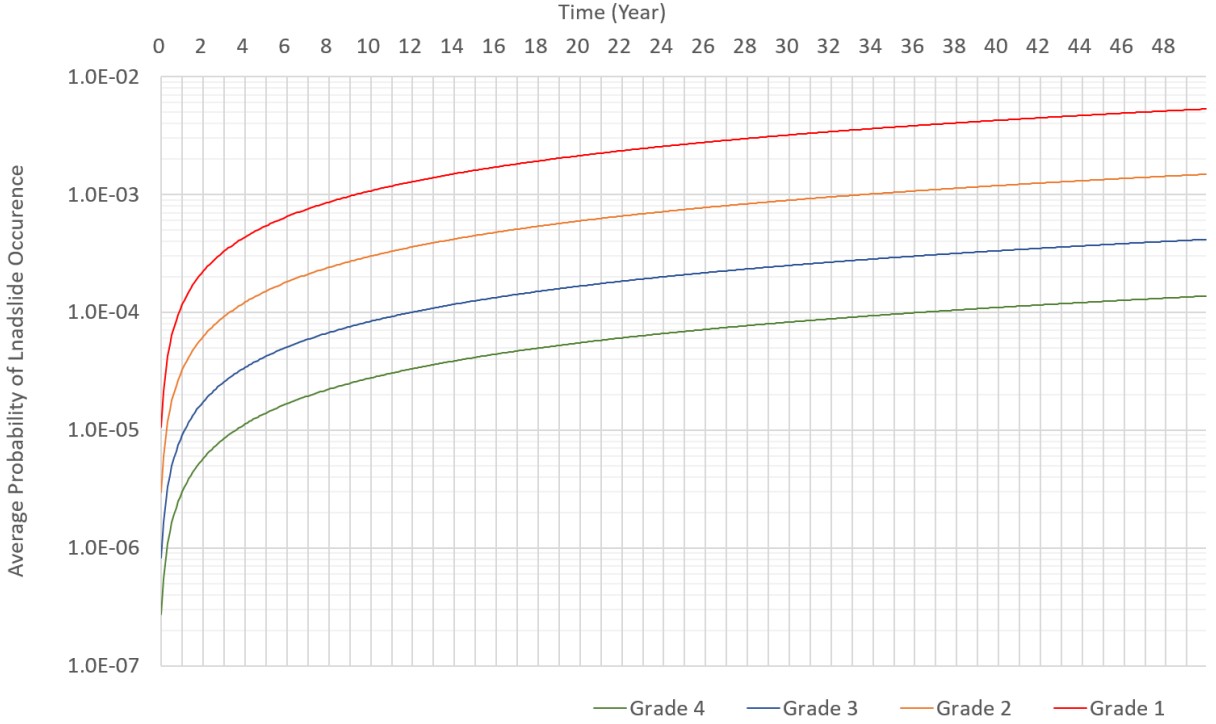

**Figure 5.** Change in average probability of landslide occurrence depending on the monitoring time interval

**Table 2.** Monitoring time interval sets producing the same level of average probability of landslide occurrence

| Landslide | Monitoring time interval (year) | | | |
|---|---|---|---|---|
| Hazard grade | 1 | 3.6 | 12.8 | 38.7 |
| 1 | 1.06.E-04 | 3.82.E-04 | 1.36.E-03 | 2.12.E-03 |
| 2 | 2.96.E-05 | 1.07.E-04 | 3.79.E-04 | 5.93.E-04 |
| 3 | 8.29.E-06 | 2.99.E-05 | 1.06.E-04 | 1.66.E-04 |
| 4 | 2.74.E-06 | 9.87.E-06 | 3.51.E-05 | 1.06.E-04 |

# 4 Discussion

## 4.1 Landslide risk management plan considering risk reduction

In the result above, we have shown that the same level of risk reduction effect can be achieved for lower landslide hazard areas with longer monitoring time intervals compared to the higher landslide hazard areas, which is valid for the landslide monitoring





methods with low temporal resolution such as those using a piezometer or inclinometer. This result can be utilized to allocate human and financial resources efficiently for preventive landslide management in a wide municipality region. In fact, the locations where landslides occur are scattered in numerous areas, especially in the mountainous region, and municipalities can find it difficult to distribute the manpower and financial budget to manage the risk that arises from landslides. Through the

analysis of the result, it is possible to comprehend that management for landslide prevention in the study region can be implemented covering 12 times more of the concerned sites for the third grade and 38 times more of the sites for the fourth grade with rotational site visiting by the same amount of resources assigned to the first grade areas, producing an equivalent risk reduction effect. The enforcement to prevent disasters from unstable slopes (2007) in South Korea requires that an authority commence monitoring inspection more than once in a year for all concerned unstable slopes; however, no scientific

judgement has been introduced to determine the appropriate monitoring time interval for risk assessment. The graph in Fig. 5 that depicts the average probability of landslide occurrence can be actively applied with revision of the frequency data that are suitable to other regions, in order to create a logical plan for landslide monitoring to cover a broad area with targeted risk reduction. In addition, if we convert the concept introduced in the probability of landslide occurrence into the probability of engineering mitigation failure such as a fence, net or retaining wall, the monitoring schedule can be planned to check their

functional competence and integrity degradation, wherever a valid frequency of flaw is recognized. In addition to this study, risk analysis can be conducted to maintain the risk below tolerable levels considering societal risk criteria (Dai et al., 2002) by implementing monitoring activities, which are for the areas with potential landslide risk that are anticipated to have loss near townships, infrastructure or ecologically valuable areas. However, this task can be performed after enough landslide inventory data are accumulated.

## 4.2 Characteristics of determining the monitoring interval with average probability

By referring to the average probability of landslide occurrence in Eq. (13), when $\lambda$ is sufficiently smaller than one ($\lambda \ll 1$), it is possible to use an approximate equation for the probability of occurrence as below (I.S.A., 2002) because the exponential $e^x$ can be simplified:


Average Probability of landslide occurrence $\approx \lambda \times TI / 2$                    (14)

Through this simplified equation, it was identified that the monitoring time intervals needed to achieve the same risk reduction effect with different landslide hazard grades are not dependent on either the landslide occurrence period estimated from the

rainfall threshold or the size of the unit area. This implies that if a municipality maintains the inventory data to count the number of landslide events according to their own hazard classification, differentiated monitoring intervals can be easily determined with decision making on risk reduction level for the highest hazard grade considering their resource pool. After reliable frequency data for landslide occurrence have been established and the extent of the scope of the area is fixed,





meaningful calculation results for risk reduction can be obtained, and mitigation measures such as monitoring to prevent loss of lives or property can be implemented in a quantitative manner founded on the individual risk or societal risk criteria. Regarding the size of the unit area, a municipality needs to decide on the management scope of the area over which to apply the frequency of landslide occurrence for landslide risk management because the frequency of occurrence is altered by the unit area. If the size of the unit area is extended from the pixel unit to the area unit of km$^2$, or to the entire county, the occurrence frequency will be increased. This means that regarding the larger unit area, the probability of a disaster occurrence will increase as the frequency of occurrence increases, and more resources will be required to reduce the risk over the scope area, which is an issue of scale related to resolving effectiveness.

## 4.3 Importance of the combination of spatial and temporal probabilities

When a landslide management system is established, it is important to combine both spatial probability and temporal probability for the system because landslides will not be initiated on high susceptibility areas with no temporal triggering factor and the distribution of the landslide events cannot be provided without spatial data (Jaiswal et al., 2010;Corominas and Moya, 2008). For landslide risk management, when a spatial unit is classified by landslide hazard grades after the analysis of susceptibility, it is necessary to collect and accumulate temporal probability data sufficiently according to the hazard grades assigning it to the spatial unit. Consequently, it is crucial to build a landslide risk management system by combining spatial and temporal information for accurate probability estimation and for a preventive mitigating action plan with a timely schedule. For this part, it is necessary to distinguish the types of landslides and adopt an appropriate unit of landslide frequency. Importantly, as a dynamic change in land surface morphology, the cause of the landslide is removed once a landslide has occurred, and it changes to a more stable condition (Guzzetti et al., 2005), so continuous collection and updating of landslide occurrence data should be accomplished following a landslide inventory framework or web-based platform in order to maintain valid frequency of occurrence data. Risk treatment is the final resolution for risk management, and a balance between simplicity for convenience and rigor in methodology should be attained to approach landslide treatment (Kadry and El Hami, 2015).

## 4.4 Preventive landslide management by periodic monitoring

As an early warning system for natural hazards can prevent unwanted loss of life or damage to property by timely detection with its designated function, and transmitting that signal to an enforcement body can result in proper mitigation (Sättele et al., 2015), and also as a preventive strategy to reinforce an earthen structure will be the most favourable option for securing the structural safety (Su et al., 2012), the preventive strategy with periodic monitoring to reduce risk is a convincing approach to landslide risk management. This approach is to reduce the likelihood of landslide occurrence, and it is a cost-efficient and preceding model in contrast to reducing the magnitude of the disasters. Implementation of landslide monitoring has the property of preventive maintenance, and it can be regarded as a leading indicator for landslide risk management. However,



the understanding of the condition by an experienced person at the local level is a pivotal factor for landslide risk management including efficiency of work because the initiating causes of landslides are complex (Nourani et al., 2014). As an example, specific and occasional landslide monitoring after extreme weather events is also essential considering seasonal factors, for instance, in the region being affected by the monsoon. Besides regular pore water pressure monitoring and deformation

monitoring, efforts should be made to increase the monitoring activities in the rainy season or to install equipment capable of wireless real-time monitoring to protect core vulnerable areas against landslides.

## 5 Conclusion

As this paper demonstrated by calculating the probability of landslide occurrence with frequency data estimated by a rainfall

threshold, an optimized set of landslide monitoring time intervals for low temporal resolution methods such as piezometer or inclinometer was analysed, searching for the same level of risk reduction effect along with different landslide hazard areas. Studying the landslide inventory data reported when Typhoon Ewiniar struck Pyeongchang County together with local weather data, the temporal probability was integrated with the spatial probability information based on the landslide hazard map classified by susceptibility differences. It was also demonstrated in this study that the scheme of reliability can be employed

to estimate the frequency of landslide occurrence with an appropriate unit and to establish the formula for the probability of landslide occurrence that is the same as the equation driven by the Poisson distribution model.

The results showed that more landslide events were initiated in the second and third hazard grade areas than the first grade area due to their total respective areas; however, the highest number of landslide events per area was observed in the first grade when we inspected the occurrence ratio by unit area. We have shown that the area with greater landslide hazard indicates the

higher value of landslide occurrence frequency in the study region, and the estimated frequency data were used for calculation of the probability of landslide occurrence. By analysing the average probability of landslide occurrence, we have ascertained that implementing monitoring with the same time interval to all concerning landslide areas will impose greater risk reduction effects to the lower landslide hazard grade areas than the higher grade areas, so that an equivalent level of risk reduction can be achieved for lower landslide hazard areas with a longer monitoring time interval.

The result proves that the timely landslide monitoring schedule according to the different landslide hazard grades can be planned by calculating the average probability of landslide occurrence considering risk reduction effects, and human and financial resources can then be allocated efficiently for preventive landslide management. The analysis of this study can be used for landslide risk management by a municipality where the areas vulnerable to landslides are scattered in a broad region, after establishing accumulated occurrence frequency data. Especially for Pyeongchang, which is not only an upcoming

Olympic host city but also a county with continuous population increase, we propose that more systematic landslide management is required to avoid potential loss due to landslides, which is a preventive strategy with a detailed plan of periodic monitoring to reduce the likelihood of landslide occurrence.



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
