# Peer review of "Risk-based analysis of monitoring time intervals for landslide prevention"

_Natural Hazards and Earth System Sciences, 2017_

## Referee Comment (RC1) · Anonymous Referee #1 · 27 Dec 2017

General Comments: The paper "Risk-based analysis of monitoring time intervals for landslide prevention" presents a landslide risk and mitigation study conducted in northern South Korea in order to establish monitoring frequencies, appropriate to reduce landslide risk over a wide area. The paper is written clearly and its method based on previous publications. By using approaches from civil engineering, Lee et al. are able to estimate the reduction in landslide probability through monitoring efforts. This combination of different risk evaluation methods will be of interest to readers of NHESS, and hence I think this paper is suitable for publication in this journal.

Nevertheless, the paper has some significant flaws that need to be addressed. My major concern is that the analysis is based on landslides triggered by one extreme weather event of 2006. Although a high number of landslides were triggered and may thus pro-

vide a good spatial estimate of landslide probability, determining a landslide frequency from a single event is, at least in my view, highly speculative. Thus some questions remain unanswered (e.g., does prolonged, intense rainfall trigger landslide, or is it only extreme events?) and I think the temporal frequency for the different hazard classes may be over- or underestimated. This, in turn, has then an impact on the calculation of the risk reduction by monitoring. Which, brings me to my second point. In your paper you state that by law risk areas have to be monitored at least once a year. In your analysis, one year is given as the minimum. Depending on the failure mechanism, some slopes may require a monitoring frequency well below one year. This is something you touch onto in your discussion, but I think this issue should play a more significantly role throughout the paper, also given that the title of the paper is "Risk based analysis of monitoring time interval to prevent landslide". Hence, it would be more appropriate to include significantly higher monitoring frequencies (hourly/daily/weekly/yearly).

Generally, I think the paper is well structured. However, the strict and comparably long description of the methodology is unnecessary. I would suggest reducing the description of the standard approaches, and extending the description of the novel risk analysis. The paper would also benefit from a revision of grammar and sentence structure by a native English speaker. There are many very long sentences, which contain an exceptional amount of information and are therefore very difficult to understand for a reader reading it for the first time. I suggest splitting those sentences in two or three separate sentences.

Specific comments:

Page 3, Line 14: You mention that topography is the main factor for landslides in the area. Although this may be the case, the geology will almost certainly play a major role as well. Hence, please add some description of the local geology (bedrock and soil cover).

Page 5, Line 14: I think developing a landslide susceptibility map on just one weather

event is not ideal. Was there no previous data available? If not, please add a discussion on the limitation of using the landslide data of one extreme weather event only and its impact on the reliability of your estimated temporal and spatial landslide probability.

Page 8, Lines 11-12: Can Typhoon Ewiniar be classified as a common example of regional weather patterns that are likely to be reoccurring, or was this a truly exceptional event? In that case, how reliable are your estimates of landslide probability?

Page 12, Lines 18-21: I think it is highly speculative to define a rainfall threshold based on a landslide inventory of just one event. Next to extreme events, are landslides triggered by prolonged, intense rainfall events, which may be more characteristic for years 2010-2014?

Page 18, Lines 5-8: Is this necessarily true? The unit is given as landslide event $\times$ pixel-1 $\times$ year-1; if the pixel size increases the probability will decrease, but with increasing size more events may be counted. Please revise this statement.

Technical comments:

Page 1, Lines 14-15: "manually read inclinometer and piezometer" – continuously logging and transmitting inclinometer and piezometer are available, but you are talking about manually logged installations.

Page 2, Line 2: "relatively expensive" is potentially a misleading expression, perhaps "comprehensive monitoring methods" may be a better choice. Page 2, Line 5: I don't think that singular is the right choice here. Although, a single inclino- and piezometer is the bare minimum, in real applications, you would have more than that. Also, could you please clarify the last two parts of the sentence.

Page 2, Line 10: "a few articles have addressed the time intervals for monitoring" – missing references.

Page 2, Lines 23-29: This is a reoccurring issue in the paper – please avoid those very long sentences, these are confusing and very difficult to follow.

Page 7, Line 6: Missing full stop after "items N"?

Figure 3: Do the red points correspond to dates with landslide occurrence?

Figure 4: I think it would be better to show landslide occurrence on the map of the hazard grades.

Figure 5: axis label should read ". . . Landslide Occurrence"

---

## Author Comment (AC1) · 15 Jan 2018

Thank you for reviewing our manuscript. Please, find our response to the comments from Referee #1 in the attached supplement.

Please also note the supplement to this comment:
https://www.nat-hazards-earth-syst-sci-discuss.net/nhess-2017-356/nhess-2017-356-AC1-supplement.zip

---

## Referee Comment (RC2) · Anonymous Referee #2 · 4 May 2018

Dear Authors, your paper "Risk-based analysis of monitoring time intervals for landslide prevention" aims at demonstrating that different monitoring time intervals would reduce the landslide risk over wide areas. In this respect, "an optimized set of landslide monitoring time intervals for low temporal resolution methods, such as piezometer or inclinometer, was analysed". The thesis at the base of the paper, as well as the title, look awkward. It is not really clear how different monitoring time intervals may reduce the level of risk. Following the risk formula provided by Varnes 1984, (R = H x V x E) the risk can be evaluated as a function of the hazard, element at risk and vulnerability. Of course in landslide prediction the monitoring phase is fundamental, but, mainly the aim is to gather information on variables responsible for landslides triggering in order to define reliable thresholds or to reduce the number of incorrect predictions in landslide

early warning systems (LEWS). Moreover the authors are analysing the occurrence of rainfall-induced landslides in a wide spread area (Pyeongchang County, South Korea), supposing to be fast slope movements (it is not clearly defined in the text), how these phenomena can be monitored using inclinometers? In this regard, it is important to have in mind the different scales of analysis we are dealing with. At a regional scale the position and sliding surface of a landslides is not known a priori. On the other hand, at a local scale, we could know the location of the landslide/s and its sliding surface. Thus, as a function of the type of landslide and scale of analysis the monitoring instruments change. Then, "to analyse monitoring time intervals for low temporal resolution . . . an equation for the probability of landslide occurrence was denoted by the concept of reliability, and the monitoring time interval was analysed quantitatively by calculating the average probability of landslide occurrence. To identity the frequency of landslide occurrence, a unit of relative temporal frequencies was adopted, and it was estimated by establishing rainfall threshold". A crucial point that needed to be clearly explained is how and why the concept of reliability can be applied to evaluate the probability of landslide occurrence. How the assumptions of the method can be generalized and applied to landslide? Which are the similarities? The same questions arise for the definition of the average probability of failure on demand. Is it possible to replace the time interval with the monitoring interval? Why? Some concerns arise for the rainfall threshold definition and the evaluation of the landslide occurrence frequency, too. For the first, the threshold is defined considering landslides triggered by one extreme rainfall event in 2006 (Typhoon Ewiniar) and the variables considered are dependent, as it possible to notice looking at figure 3. Using different variables would have helped. Moreover the threshold defined by the authors (150 mm for daily precipitation and 200 mm for 48-hours) have been exceeded 3 times in 11 years. But it seems that only 1 out of 3 corresponded to landslides occurrence (july 2006, Typhoon Ewiniar), it means that the threshold had 66% of false alerts, which is a quite high error. Then, the probabilistic period of landslide has been evaluated but it is not clearly explained how. However, it looks more as a return period of threshold exceedances than a probabilistic period.

Concerning the landslide occurrence frequency in table 1, the values seem to be incorrect. However, it the dimension of a spatial distribution of landslides than a frequency of landslide per each hazard zone. I would have expected a rate of the number of landslides occurred in each hazard zone and the total, in the period of analysis (11 years?). Finally, the authors state that different monitoring time intervals reduce the average probability of landslide (Table 2). The table proposes 4 different time intervals $(1-3, 6-12, 8-38, 7$ years) which seem to be really too wide to me. Depending on the type of landslide, some slopes may require a monitoring frequency well below one year. Moreover the statement "we have shown that the same level of risk reduction effect can be achieved for lower landslide hazard areas with longer monitoring time intervals compared to the higher landslide hazard areas", reads a bit counter-intuitive. Other major issues are represented by the structure of the paper which results confused. A more linear structure would be: Introduction. 2 Method. 3 Case study. 4 Result. 5 Discussion. The English is difficult to follow, also because the sentences are often too long and not-well articulated. Finally the topic of the paper not really fit the aim of the Special Issue. My overall judge is that the assumptions are not well explained and justified and the paper fails in its crucial parts. For the above mentioned reasons, even if I acknowledge the effort in producing the manuscript, from my point of view the paper cannot continue the publishing process, and a fully reconsideration is needed before a new submission.

Best regards